# Designing of an Oat-Mango Molded Snack with Feasible Nutritional and Nutraceutical Properties

**DOI:** 10.3390/foods13213402

**Published:** 2024-10-25

**Authors:** Yudit Aimee Aviles-Rivera, José Benigno Valdez-Torres, Juan Pedro Campos-Sauceda, José Basilio Heredia, Jeny Hinojosa-Gómez, María Dolores Muy-Rangel

**Affiliations:** 1Centro de Investigación en Alimentación y Desarrollo, A. C., Subsede Culiacán, Carretera Eldorado Km 5.5, Campo el Diez, Culiacán CP 80110, Sinaloa, Mexico; yaviles221@estudiantes.ciad.mx (Y.A.A.-R.); jvaldez@ciad.mx (J.B.V.-T.); jbheredia@ciad.mx (J.B.H.); 2Tecnológico Nacional de México: Instituto Tecnológico de Culiacán, Juan de Dios Bátiz No. 310 pte. Col. Guadalupe, Culiacán CP 80220, Sinaloa, Mexico; juan.cs@culiacan.tecnm.mx; 3CONAHCYT-Centro de Investigación en Alimentación y Desarrollo, A. C., Subsede Culiacán, Carretera Eldorado Km 5.5, Campo el Diez, Culiacán CP 80110, Sinaloa, Mexico; jeny.hinojosa@ciad.mx

**Keywords:** food design, functional foods, *Mangifera indica*, properties, quality

## Abstract

In recent years, the market has seen a growing demand for healthy and convenient food options, such as fruit and cereal bars, driven by shifts in eating habits. These changes are primarily attributed to time constraints in meal preparation and the need for ready-to-eat foods. Consequently, this has promoted interest in creating a nutritious, high-quality snack combining oats and mango. This study employed a response surface analysis of extreme vertex mixtures, incorporating constraints and three components: oats, mango peel, and dehydrated mango pulp. This resulted in ten different mixtures, each with unique combinations and proportions of the three components. It evaluated the microbiological quality, proximal composition, total phenolic content, tannins, Aw, color, texture, and chemical properties during storage at room temperature. The optimal blend, which demonstrated the best quality characteristics, consisted of 44.38% oats, 5.36% mango peel, and 29.24% mango pulp. This formulation yielded a protein content of 7.1 g, dietary fiber of 20.3 g per 100 g, total phenols of 3.4 mg gallic acid per g, and no pathogenic microorganisms. According to the obtained data, Aw > 0.3, the estimated shelf life could be 12 months at room temperature. Developing a stable oat-mango snack with excellent nutritional, nutraceutical, chemical quality, and microbiological properties is technologically feasible.

## 1. Introduction

Mango (*Mangifera indica* L.) is a delicious tropical fruit and a nutritional powerhouse, making it one of the most sought-after fruits in the global market. Its sensory, nutritional, and functional properties have economically secured its position as the third most important fruit after bananas and oranges [1,2]. In 2022, global production of mango, guava, and mangosteen fruits reached 59.15 million tons, with Mexico ranking fifth with 1.8 mt, following India, China, Indonesia, and Thailand [3].

The “Ataulfo” variety is native to Mexico, and its widespread commercialization worldwide is attributed to its distinctive flavor, firm texture, sweet flesh, low acidity, and intense aroma [1,4]. Due to these appealing characteristics, a portion of the production is allocated for processed foods, including juices, nectars, jams, and canned and dehydrated products [5]. However, this industrial processing generates up to 60% of waste, primarily peel, seed, and fiber, leading to economic losses, environmental challenges, and possible uses [6].

Over the past two decades, research has focused on characterizing and processing mango peels for their potential reuse in food development. This not only offers a new source of income for the mango processing industry [5,6] but also emphasizes the nutritional value of this by-product. Mango peels are rich in dietary fiber, polyphenols, proteins, and carotenoids, making them a promising source of functional ingredients [6,7,8,9].

Mango by-products have been utilized in producing various processed foods, including fermented beverages, jellies, chips, pasta, cookies, bread, and snacks [10,11,12,13]. The results demonstrate that these products are rich in phenolic compounds, exhibit antioxidant activity, possess probiotic potential, and are high in dietary fiber. Consuming products containing these compounds is associated with several health benefits. For instance, a high dietary fiber intake is crucial in addressing obesity, coronary heart disease, cancer, and diabetes problems [14,15,16]. Additionally, the bioactive compounds in these products play a significant role by neutralizing free radicals, thus preventing cellular damage [17]. This information is essential for understanding the health benefits of these products.

The competitive landscape of the food industry continually drives the search for innovative product designs [18]. In recent years, there has been a growing demand for healthy foods that are convenient and sensorially appealing. As a result, fruit and protein bars, along with a variety of other healthy snacks, are gaining popularity. This research focuses on developing and optimizing a snack made from oats, mango pulp, and mango peel flour, offering substantial benefits in terms of nutritional value, nutraceutical properties, and quality stability at room temperature. This presents a promising opportunity for the food industry.

## 2. Materials and Methods

### 2.1. Materials

In this study, ripe mangos of the Ataulfo variety, cultivated in the South Pacific region of Chiapas, México, were used. Mango fruits were washed with water and sanitized with 20 ppm sodium hypochlorite for 15 min. Once sanitized, the fruits were peeled, cut into pieces, and crushed in a T12L stainless steel industrial blender (Rbanda, Mexico) to a homogeneous paste and stored at −17 °C. The fruit peels were placed in a stainless-steel strainer and blanched in water at 90 °C for 120 s. To inactivate the enzymes responsible for deterioration reactions such as browning, they were cut and dried for 20 h at 50 °C ± 2 °C in a hot air dehydrator (Excalibur Food Dehydrator, Burleigh Heads, QLD, Australia) until reaching a constant weight. The peel was pulverized in a mill (Pulvex 200) and stored at 2 °C. Commercial oat flakes Granvita (Grupo Industrial Vida S.A. de C.V.) were used, which contained protein = 10%, fat = 10%, total carbohydrates = 70%, and dietary fiber = 13.3% on dry matter.

### 2.2. Snack Design, Preparation, and Characterization

All mango-oat combinations described in Table 1 were added to 21% of the complement (15% sucrose, 5.3% butter, 0.5% salt, and 0.2% baking powder) to achieve 100%. All components were mixed to make a homogeneous mass, which was molded into an 8 cm long and 7 mm thick indented cylindrical shape. Each sample was left to rest for 15 min at room temperature and baked at 180 °C for 6 min, then allowed to rest for 8 min at room temperature (to allow the water in the product to migrate from the inside to the outside to reach equilibrium), followed by a second cooking stage of 10 min at 180 °C. The samples were removed from the oven, cooled to room temperature, and meticulously packaged in Ziploc^®^ plastic bags for further analysis, ensuring the product’s quality.

### 2.3. Proximal Analysis

Moisture, ash, protein (N × 5.83), fat, and carbohydrates (by difference) contents were determined according to AOAC methods [19]. Total dietary fiber was evaluated by the Megazyme total dietary fiber assay procedure, which is based on the AOAC 991.43 method (AOAC, 1998).

### 2.4. Hardness

The hardness of the samples was determined using a TA-XT2i texturometer (AMETEX, Cassatt Road Berwyn, FL, USA) measured at 2 points on the surface of the product with a double blade attachment at a speed of 20 mm/s and a contact force of 1 Newton until total breakage of the sample structure [20]. The results were reported in Newton (N).

### 2.5. Color Analysis

The color attributes of the samples were obtained in the CIELCh color space using a spectrophotometer (CM-700d; Konica Minolta, Ramsey, NJ, USA). The luminosity (L), hue angle (Hue = arc tan b/a), and chromaticity (Chroma = √ [a^2^ + b^2^]) were obtained with OnColor QC version 5 (CyberChrome, Stone Ridge, NY, USA) [21,22].

### 2.6. Chemical Analysis

The samples’ pH, acidity, and total soluble solids (TSS) were carried out according to [19]. To 10 g of sample, 50 mL of distilled H_2_O pH 7.0 was added, liquefied, and filtered. A total of 50 mL of filtered sample was taken and placed in a plastic cup. The pH and percentage of acidity were determined with a Mettler Toledo T50 automatic titrator using the LabX program, where the dilution factor and the milliequivalent of citric acid were indicated. One milliliter of the sample was placed in the Metter Toledo S470 refractometer cavity for total soluble solids. The result was multiplied by the dilution factor and reported in degrees Brix (°Brix).

### 2.7. Antioxidant Capacity

Extract preparation for antioxidant capacity involved 2 g of dry sample being added to 10 mL methanol, shaken at 200 rpm for 2 h, and centrifuged at 27,670× *g* for 20 min at 4 °C [23], using a centrifuge mod Z36 HK (Hermle Labortechnik, Wehigen, Germany).

The Folin Ciocalteu method was used to quantify the total phenolic compounds. A total of 10 µL of the extract, 230 µL of distilled water, and 10 µL of the 2 N Folin Ciocalteu reagent were placed in a clear flat-bottom 96-well transparent plate, incubated for 3 min, 25 µL of Na_2_CO_3_ 4 N was added, and incubated for 2 h in the dark. The absorbance was measured at 725 nm using a Synergy HT Microplate Reader (BioTek, Instruments, Inc., Winooski, VT, USA). The results were expressed as mg of gallic acid equivalents (GAE) per gram [23].

The antioxidant capacity was determined by the DPPH (2,2-diphenyl-1-picrylhydrazyl) method. A 20 µL sample extract was prepared and added to the microplate. A 20 µL sample extract, using Trolox (0, 0.05, 0.1, 0.2, 0.25, 0.3, 0.35, 0.4, 0.5, 0.6, 0.8, and 1.0 mg∙mL^−1^); and 200 µL of DPPH (2,2-definil-1-picrylhydrazyl radical) reagent were added and incubated at 20 °C for 30 min, under dark conditions to determine the calibration curve. Absorbance was measured at 540 nm using a Synergy microplate reader HT (BioTek, Winooski, VT, USA). The results were expressed as mg Trolox equivalents/100 g sample dry weight [24].

### 2.8. Total Tannins

Total tannins were obtained as the difference between the content of phenolic compounds and the non-tannin phenolics compounds. First, it was necessary to determine the content of phenolic compounds using the Folin Ciocalteu assay, and then the content of non-tannin phenolics was subtracted from that value. For this, tannin precipitation was performed with PVPP. A total of 100 mg of polyvinyl polypyrrolidone (PVPP) and 1 mL of extract were added to Eppendorf tubes, shaken in a Vortex Maximix LP (Thermo Scientific, Waltham, MA, USA), and incubated for 15 min at 4 °C. Then, they were shaken rapidly and centrifuged for 10 min at 4427× *g*. The supernatant was recovered, and the content of phenolic compounds (non-tannin compounds) was determined using the Folin Ciocalteu assay. The results were expressed as mg catechin equivalents (CE)/g [25,26]. Total tannins = (the content of phenolic compounds minus the non-tannin phenolics compounds).

### 2.9. Water Activity

A 2 g mango bar was placed in a cell in the equipment cavity of Aqua LabCX-2 (Cromer, Australia) dew point hygrometer at 25 °C, previously calibrated with a NaCl standard of 0.75 water activity (Aw). Finally, Aw was measured and recorded in the sample.

### 2.10. Microbiological Analysis

*Escherichia coli* (*E. coli*), fecal coliforms, and total coliforms were determined following the Mexican Official Standard NOM-210-SSA1-2014 [27], appendix H, to *Salmonella* spp. appendix A and appendix B, the procedure for food. A sample of 25 g was placed in a sterile bag, 225 mL of phosphate butter was added, and the sample was homogenized manually for 2 min (stock sample). Coliforms from this homogenate were analyzed with decimal dilutions made using 9 mL of phosphate buffer. Biochemical tests were performed to evaluate *E. coli*, and the results were reported as the most probable number/mL (MPN/g) for *Salmonella* expressed as presence-absence in 25 g of sample. The fungi and yeasts were analyzed according to the Mexican Official Standard NOM-111-SSA1-1994 [28] for products intended for human consumption by plate count.

### 2.11. Data Analysis

A mixture design of degree 2, with extreme vertices, was implemented with three components (oats, mango peel flour, and mango pulp) with the following constraints: oats 30–55%, mango peel flour 0–30%, and mango pulp 25–65%. Oats and mango (peeled flour and pulp) constituted a balanced 79% of the total mixture, and 21% was the complement in the formulation (15% sucrose, 5.3% butter, 0.5% salt, and 0.2% baking powder). The mixture design consisted of 10 mixtures [29] and each mixture was replicated thrice (Table 1). Statistical significance of coefficients for the terms in the models were taken as *p* < 0.05.

## 3. Results and Discussion

### 3.1. Nutritional Characterization of Snack Ingredients

Table 2 shows the nutritional and nutraceutical content of the components used to prepare the oat-mango snack. The moisture content of the mango pulp, at 79%, served as the primary water source in the mixture, facilitating the kneading process with the other ingredients. The nutritional and nutraceutical characteristics of the snack were meticulously estimated, considering the protein contribution from oats (10%), dietary fiber (21.5%), and primarily tannins from mango peel. These estimations provide a solid basis for confidence in the snack’s health benefits. The raw materials’ analyzed content was consistent with values reported by various authors for oat and mango fruits and peels [5,7,8].

### 3.2. Optimization Process

Table 3 presents the protein, dietary fiber, and bioactive compound contents across the ten treatments derived from the experimental design, which were influenced by the proportions of oats, mango peel, and pulp. Higher ratios of oats and mango peel flour in the combinations enhanced the protein and dietary fiber content. Conversely, including mango pulp and mango peel flour improved the total phenol values, along with the incorporation of tannins. It was reported that mango peel is rich in total phenols, which help eliminate free radicals and function as natural antioxidants [8]. These findings offer a significative promise for future food development, opening opportunities for the creation of healthier and nutritious snack.

The dietary fiber functionality of oat-mango snacks, determined by the content and ratio of soluble and insoluble fractions [30], significantly influences glucose levels. The soluble fiber in these snacks helps stabilize glucose levels, providing health benefits. Furthermore, soluble fiber serves as a prebiotic, nourishing beneficial gut bacteria, which promotes a balanced microbiota and can enhance overall consumer health [16,31]. Furthermore, consuming foods that contain tannins could be therapeutic; they prevent cancer and the formation of carcinogens. Similarly, tannins enhance body’s natural detoxification defenses [25].

### 3.3. Regression Models

The ANOVA for the response variable is shown in Table 4. Protein and dietary fiber were adjusted by second-order models, while total phenols and total tannins were adjusted by linear models. The significant regression coefficients (*p* < 0.05) for each response are shown in Table 5. The corresponding polynomial models and their goodness of fit (R^2^) are given in Table 6.

### 3.4. Optimization

Using Minitab 19, an overlap of the contour plots of all response models over the restricted experimental region showed a small area where all responses intersect (Figure 1). A feasible solution in this region was oat = 0.441 (44.1%), mango peel flour = 0.0648 (6.5%), and mango pulp = 0.284 (28.4%). The response values at this point were protein (%) = 6.12, dietary fiber = 23.45 (%), total phenols (mg GAE∙g^−1^ snack) = 3.67, and total tannins mg Catechin∙g^−1^ snack = 2.89. This result was corroborated by the optimization procedure based on the optimal desirability functions for the four desired variables of 0.805.

### 3.5. Physicochemical Properties of the Optimized Mango-Oat Snack

The snack made with the feasible oat-mango blend demonstrated important protein, dietary fiber, total phenols, and tannin contributions (Table 7), which were consistent with the results predicted by the model (Figure 1), indicating excellent reproducibility of the experiment. Overall, the optimized snack contained 3.8% moisture (Table 8), which falls within the acceptable range for baked goods according to NOM-247-SSA1-2008 [32] (moisture < 8%). The protein content is primarily derived from oats, followed by mango peel. With a high dietary fiber content of 20.3% (with a ratio of 3:1 insoluble to soluble fiber), this snack has three times the fiber found in other snacks enhanced with mango peel [4,33,34], potentially offering significant health benefits.

A serving of baked cereal products with fruit is 40 g by regulations (Grain-based bars with or without filling or coating, e.g., breakfast bars, granola bars, rice cereal bars, piece or g [35] from USA) and NOM-086-SSA1-1994 [36] from México. In this context, a serving of the oat-mango snack (40 g) provides 8.1 g of dietary fiber, which accounts for 29% of the recommended daily intake (DI), where 20% DI or more of a nutrient per serving is considered high from USA [35], qualifying it as “high in dietary fiber”. The insoluble dietary fiber in the snack primarily comes from the mango peel, while the soluble dietary fiber is derived mainly from the oat content [37]. The nutritional properties and quality of fiber are influenced by the composition and ratio of soluble and insoluble fractions; oats contain over 50% soluble fiber [38], whereas mango peel and pulp are richer in insoluble fiber [39]. However, a 40 g serving of the oat-mango snack contains 6.5 g of sugars, representing 26% of the daily value recommended by the World Health Organization (maximum 25 g/days). This suggests that sensory studies should be conducted to reduce the 15% added sugar content and assess its impact on acceptability.

Fiber-rich by-products such as oatmeal and mango can be incorporated into food products as substitutes for flour, fat, or sugar, enhancing water and oil retention and improving emulsion and oxidative stability [38]. Dietary fiber plays a crucial role in the body, it helps reduce glucose absorption in the bloodstream, leading to better insulin control and maintaining a healthy intestinal ecosystem, among other benefits [16]. The consumption of foods with soluble dietary fiber (SDF) to a total dietary fiber (TDF) ratio of 0.5 is associated with these positive characteristics [40]. In the optimized oat-mango snack, this ratio was 0.22, indicating a higher contribution of insoluble dietary fiber (Table 8).

In recent years, the food industry has increasingly focused on developing products that contain bioactive ingredients, utilizing by-products from fruit processing and marketing them as functional foods. These foods offer health benefits beyond basic nutrition, often due to the inclusion of bioactive compounds. Our study found that the oat-mango snack has a phenolic content of 3.4 mg of GAE per gram (Table 7), comparable to the 5.3 mg of GAE per gram reported by [4] for bread made with mango peel flour.

Mango peel, a significant contributor to our oat and mango snacks, is rich in antioxidant phenolics and tannin compounds. Tannin incorporation was meticulously considered in our mixture combinations, resulting in a snack with beneficial properties that boasted 2.2 mg of catechin per gram (Table 7). Although specific values for tannin consumption are unavailable, it is generally recommended not to exceed 2.5 g per day [41].

The antioxidant capacity of the optimal oat and mango snack was 41.4 µmol ET∙g^−1^ (Table 8), this value is in a range of results found in various foods [33,34,42]. It is important to note that the antioxidant capacity of a product is influenced not only by the content of phenolic compounds but also by the presence and levels of other bioactive compounds [40].

The optimized oat-mango snack offers benefits with its excellent physical, chemical, and physicochemical qualities (Table 8). Products with water activity (Aw) values less than 0.3, such as our snack, exhibit superior stability due to the low availability of water, which inhibits microbial growth and reduces biochemical changes during storage and distribution [43]. The low Aw value in the oat-mango snack was achieved using mango peel, oats, and dehydrated ingredients, with only the moisture from the mango pulp being added. Additionally, the snack achieved 22°Brix and 60.6 N, values comparable to those reported for cookies made with 20% mango peel flour [44].

The color of food plays a crucial role in its marketing and consumption. An appealing color can enhance a product’s acceptability, often independent of its flavor and nutritional content [45]. The oat-mango snacks exhibit orange hues (Figure 2), characteristic of both the pulp and peel of the mango [46,47]. These snacks display luminosity values of 57.2, chromaticity of 41.6, and a hue angle of 79.1, indicating that the optimized snack appears dark orange due to the heat treatment during baking at 180 °C. In contrast, Ataulfo mango pulp, fresh ripe, has luminosity values of 60.6, chromaticity of 63.4, and a hue angle of 62.9 [44]. Ref. [47] reported a luminosity of 51.5 and a hue angle of 55.7 for Ataulfo mango pulp powder. The color values for the oat-mango snack align closely with those of the Ataulfo mango, falling within similar orange ranges. The combination of the ingredients and the baking process results in lower color saturation and a slight product darkening, as reflected in the reduced luminosity values.

### 3.6. Microbiological

The feasible oat-mango snack demonstrated acceptable microbiological quality, attributed to the high quality of the ingredients and the baking process conducted at 180 °C. No presence of *Escherichia coli*, total coliforms, fecal coliforms, fungi, or yeasts was detected. Additionally, *Salmonella* spp. was absent in 25 g of the sample, and only 50 colony-forming units of aerobic mesophiles were found per gram, significantly below the limit of 30,000 CFU·g^−1^ established for cookies according to NMX-F-006-1983 [48]. Furthermore, the aerobic mesophile counts are acceptable for cereal-based foods, edible seeds, flours, semolina, mixtures, and bakery products NOM-247-SSA1-2008 [32].

## 4. Conclusions

From a technological standpoint, the oat and mango snacks offer acceptable nutritional and nutraceutical benefits and low water activity (Aw) values that enhance stability. The optimal formulation identified through the optimization process for the specified study region consists of 44.1% oats, 6.5% mango peel, and 28.5% mango pulp. The mixture yields a snack rich in protein, high in dietary fiber, and contains significant phenolic compounds; however, adjustment of the added sugars is necessary. Ultimately, it successfully elaborates a snack that is microbiologically safe, nutritious, easy to handle, visually appealing, and capable of maintaining its shape during distribution. The stability of this product ensures a longer shelf life, making it an attractive option for food product developers, including food design and nutrition innovation. Moreover, this approach opens new avenues for utilizing mango peel as a valuable ingredient in fortified foods for human consumption. However, it is crucial to continue research on suitable packaging and shelf life for the mango oat snack to ensure it retains its nutritional and nutraceutical quality as much as possible.

## Figures and Tables

**Figure 1 foods-13-03402-f001:**
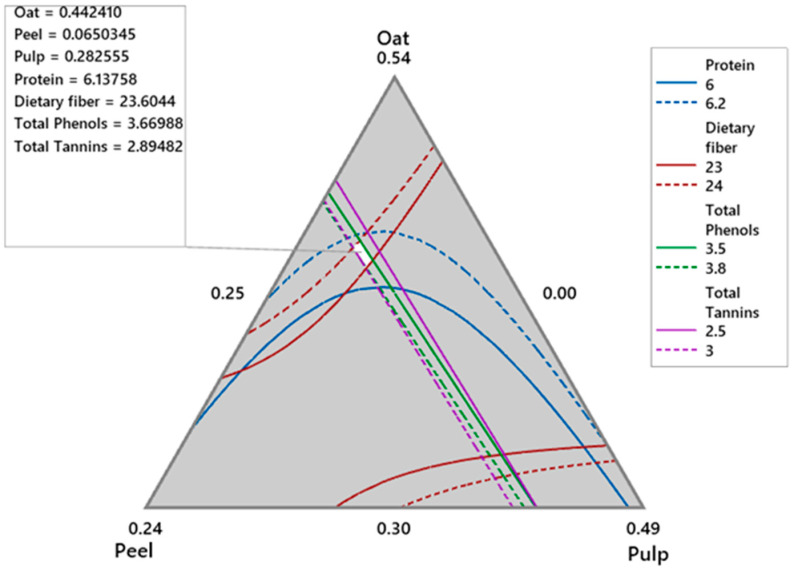
Overlaid contours of response variables of the oat-mango snack.

**Figure 2 foods-13-03402-f002:**
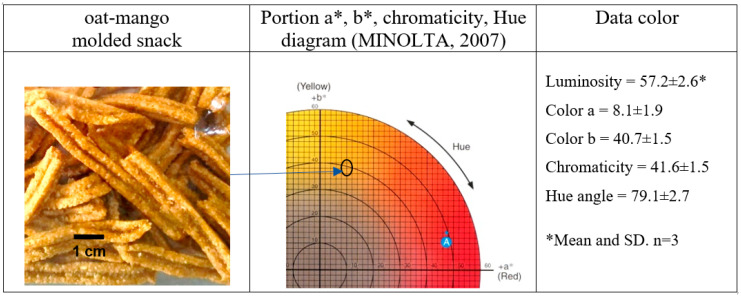
Picture and the color variables in the oat-mango snack.

**Table 1 foods-13-03402-t001:** Experimental runs, primary components (proportions), and experimental region of the extreme vertices design.

Treatments	Oat Flakes	Mango Peel Flour	Mango Pulp	Experimental Region
1	0.30	0.00	0.49	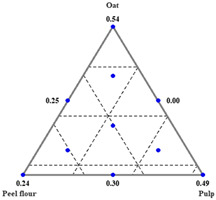
2	0.54	0.00	0.25	
3	0.30	0.24	0.25	
4	0.30	0.12	0.37	
5	0.42	0.00	0.37	
6	0.42	0.12	0.25	
7	0.38	0.08	0.33	
8	0.34	0.04	0.41	
9	0.46	0.04	0.29	
10	0.34	0.16	0.29	

**Table 2 foods-13-03402-t002:** Characteristics of primary components for oat-mango snacks.

Components	Mango Pulp (FM)	Mango Peel Flour (DM)	Oat Flakes (DM)
Moisture, %	79.3 ± 0.1 *	8.3 ± 0.28	4.5
Protein, %	1.2 ± 0.1	3.8 ± 0.13	10.0
Dietary fiber, %	3.12 ± 0.8	21.50 ± 1.6	13.3
Total phenols, mg GAE∙g^−1^	1.6 ± 0.05	127.3 ± 0.25	NR
Total tannins, mg CAT∙g^−1^	0.05 ± 0.001	50.58 ± 1.2	NR

* Mean and standard deviation of three replicates. GAE: gallic acid equivalents. CAT: Catechin. NR. Not reported. FM: Fresh matter. DM: Dry matter.

**Table 3 foods-13-03402-t003:** Means ± Standard deviation of the response variables of 10 mixtures of oats-mango snacks.

Mixtures			Response Variables			
Oat	Peel	Pulp	Protein (%)	Dietary Fiber (%)	Total Phenols (**)	Total Tannins (***)
0.30	0.00	0.49	5.7 ± 0.4 *	29.1 ± 0.4	1.0 ± 0.1	0.2 ± 0.05
0.54	0.00	0.25	6.4 ± 0.1	31.7 ± 0.1	0.7 ± 0.01	0.1 ± 0.01
0.30	0.24	0.25	6.0 ± 0.9	23.1 ± 0.6	14.2 ± 0.4	10.6 ± 0.1
0.30	0.12	0.37	4.6 ± 0.2	21.1 ± 0.2	6.6 ± 0.2	5.3 ± 0.1
0.42	0.00	0.37	6.3 ± 0.1	19.4 ± 0.1	0.2 ± 0.1	0.1 ± 0.01
0.42	0.12	0.25	6.5 ± 0.2	23.6 ± 0.2	6.0 ± 0.2	5.0 ± 0.2
0.38	0.08	0.33	5.5 ± 0.5	22.3 ± 0.5	6.2 ± 0.1	4.0 ± 0.3
0.34	0.04	0.41	6.7 ± 0.1	22.3 ± 0.5	2.3 ± 0.2	2.0 ± 0.3
0.46	0.04	0.29	6.2 ± 0.6	20.7 ± 0.6	2.0 ± 0.1	1.5 ± 0.2
0.34	0.16	0.29	5.1 ± 0.1	20.6 ± 0.1	8.4 ± 0.4	7.5 ± 0.2

* Mean and standard deviation of three replicates, ** (mg GAE∙g^−1^ snack), *** (mg CAT∙g^−1^ snack). GAE: gallic acid equivalents. CAT: Catechin.

**Table 4 foods-13-03402-t004:** Analysis of variance for protein, dietary fiber, total phenols, and total tannins.

Protein (%)						
Source	DF	Seq SS	Adj SS	Adj MS	F-Value	*p*-Value
Regression	3	2.354	2.354	0.7846	2.70	0.013
Linear	2	1.191	1.352	0.6762	2.41	0.017
Quadratic	1	1.163	1.163	1.1627	4.14	0.08
Peel * Pulp	1	1.163	1.163	1.1627	4.14	0.08
Residual Error	6	1.686	1.986	0.2010		
Total	9	4.040				
Dietary fiber (%)						
Regression	3	106.72	106.72	35.573	6.35	0.027
Linear	2	20.61	101.87	50.936	9.10	0.015
Quadratic	1	86.11	86.11	86.109	15.38	0.008
Oat * Pulp	1	86.11	86.11	86.109	15.38	0.008
Residual Error	6	33.59	33.59	5.598		
Total	9	140.31				
Total phenols (mg gallic acid equivalents∙g^−1^ snack)						
Regression	2	172.498	172.498	86.2492	125.15	0.000
Linear	2	172.498	172.498	86.2492	125.15	0.000
Residual Error	7	4.824	4.824	0.6892		
Total	9	177.323				
Total tannins (mg catechin∙g^−1^ snack)						
Regression	2	106.550	106.550	53.2749	689.54	0.000
Linear	2	106.550	106.550	53.2749	689.54	0.000
Residual Error	7	0.541	0.541	0.0773		
Total	9	107.091				

DF = Degree of Freedom; Seq SS = Sequential sums of squares; Adj SS = Adjusted sums of squares; Adj MS = Adjusted mean squares. * = Interaction effect.

**Table 5 foods-13-03402-t005:** Estimated regression coefficients.

	Response Variables			
Terms	Protein (%)	Dietary Fiber (%)	Total Phenols *	Total Tannins **
Oat	8.82	151.561	−0.5102	−0.4845
Mango peel flour	26.84	−68.235	55.1952	43.0926
Mango pulp	7.15	178.793	1.5076	0.6740
Mango peel flour * Mango pulp	−83.21	--	--	--
Oat * Mango pulp	--	−716.065	--	--

* (mg GAE∙g^−1^ snack), ** (mg CAT∙g^−1^ snack). GAE: gallic acid equivalents. CAT: Catechin. Mango peel flour.

**Table 6 foods-13-03402-t006:** Regression models and coefficient of determination.

Response Variable	Models	R^2^ (%)
Protein (%)	8.82 O + 26.84 Pe + 7.15 Pu − 83.21 Pe * Pu	58.3
Dietary fiber (%)	151.56 O − 68.24 Pe + 178.79 Pu − 716.07 O * Pu	76.1
Total phenols *	– 0.51 O + 55.20 Pe + 1.51 Pu	97.3
Total tannins **	– 0.48 O + 43.09 Pe + 0.67 Pu	99.5

* (mg GAE∙g^−1^ snack), ** (mg CAT∙g^−1^ snack). GAE: Gallic acid equivalents. CAT: Catechin. O = Oat, Pe = Mango peel flour, Pu = Mango pulp.

**Table 7 foods-13-03402-t007:** Results for predicted and experimental values in the feasible mixture of dehydrated mango and walnuts for conditions of 60 °C and 3 min.

Values	Response Variables			
	Protein (%)	Total Dietary Fiber (TDF, %)	Total Phenols (**)	Total Tannins (***)
Predicted by model	6.1252	23.4483	3.6687	2.891
Experimental data	6.6 ± 0.6 *	20.3 ± 0.4	3.4 ± 0.1	2.2 ± 0.1

* Mean and standard deviation of three replicates, ** (mg GAE∙g^−1^ snack), *** (mg CAT∙g^−1^ snack). GAE: gallic acid equivalents. CAT: Catechin.

**Table 8 foods-13-03402-t008:** Quality characteristics of the optimized oat-mango snack expressed on dry matter.

Nutritional/Nutraceutical Quality		Physicochemical Characteristics	
Moisture	* 3.82 ± 0.2	Aw	* 0.31 ± 0.002
Fat	7.4 ± 0.6	Texture (N)	60.6 ± 2.6
Ash	2.3 ± 0.2	pH	4.8 ± 0.02
Total carbohydrates	79.8 ± 0.3	TSS (°Brix)	22.0 ± 1.0
Sugars	18.9 ± 0.7	Titratable acidity (% citric acid)	0.08 ± 0.003
Insoluble dietary fiber (IDF), %	15.2 ± 0.6		
Soluble dietary fiber (SDF), %	5.1 ± 0.2		
SDF/TDF ratio	0.22 ± 0.3		
DPPH antioxidant capacity, µmol Trolox·g^−1^	41.4 ± 0.5		

* Mean and standard deviation of three repetitions. TDF: Total dietary fiber (%, Table 5). DPPH = 2,2-diphenyl-1-picrylhydrazyl. Aw: water activity. TSS: total soluble solids. N: Newtons.

## Data Availability

All data generated or analyzed during this study are included in this published article.

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
