# Peer review of "Designing of an Oat-Mango Molded Snack with Feasible Nutritional and Nutraceutical Properties"

_foods, 2024, doi:10.3390/foods13213402_

Round 1

Reviewer 1 Report

Comments and Suggestions for Authors

line 59-63 : incorrect sentence, need to be more consice

line 90 Kjeldahl factor 6,25. As for cereals this is not correct !! 

line 107 - specify organza cloth 

line 115 - sentence not correct

line 116 - rpm is related to the brand and type of centrifuge, better express in g (relative centrifugal force)

table 1 : percentages -> please use % and not 0,.. Use the same expressio of ratio throughout the complete text and tables

line 190-193 be more concise

table 3 - composition on dry matter or as eaten ?

see remarks on water content. as Mango pulp is the only ingredient which is high in moisture it has a big impact on the baked good composition. If the pulp increases, you will have more water to be removes during baking. Also the sugar content will be highet in these products. I miss this critical view throughout the discussion.

Table 8: it would be nice to make a difference between sugars and other carbohydrates as you promote the final snack as healthy.

line 233 which is the regulation stating that 1 serving is 40g ?and high in dietary fibre. mention the regulation more properly (in relation to labeling of claims within a certain region, country). As it is written now, it is not clear. 

please in the conclusion include the (acceptable) sugar content as it is (8 g per serving ?)

Author Response

Thank you for all your contributions. Attached is a document highlighting in red the points that have been addressed.

I especially appreciate your thorough review of the document. There are often details that can be overlooked or undervalued, such as sugars, the factors for protein quantification, the proximals result, and others.

Reviewer 2 Report

Comments and Suggestions for Authors

The comments are shown in the uploaded pdf.

Author Response

Thank you for all your contributions. Attached is a document highlighting the points that have been addressed in red. I especially appreciate your  review of the document. 

Sorry. The methodology for tannins used is approved and has been published in several articles. This was the one we standardized, and it gave the expected results. We consider it to be a good methodology. Thank you.

Reviewer 3 Report

Comments and Suggestions for Authors

General comments:  The manuscript describes the optimization and production of a ready to eat snack product produced from commercial oat flakes and dehydrated peels from mango fruits.  The project serves as an example of value-added uses of food waste materials.  The manuscript is well written and follows the instructions to the authors with only some minor corrections needed. The authors need to be aware of using subjective words such as “good” and “excellent”.  There are regulatory levels for such descriptors, in addition, the manuscript needs to read like a scientific study, not a sales advertisement.

Line 72:  Change “until” to “to”

Line 73:  Change “was” to “and”

Lines 114 to 115:  Remove the whole line and start the paragraph with “Two g of dried sample were shaken with 10 mL methanol at 200 rpm…”

Line 117:  Remove “Total phenols”

Lines 118 to 119:  Change to read “Ten µL of the methanolic extract was diluted with 230 µL of distilled water.  Ten µL of 2 N Folin…”

Line 125:  Remove “Antioxidant capacity”

Section 2.8:  Is there not a standard method for the determination of total tannins?

Line 146:  Change to read “A 2 g sample of the mango…”

Line 165:  End the sentence with “…in Table 1.” and start the next with “Each mixture…”

Line 218:  How is “good” defined here?

Line 275 and following:  This is a good point to make.

Line 288:  Change “excellent” to “acceptable”

Line 298:  Change “excellent" to “valuable”

Line 327:  Change “J. Sci. Food Agric.” to “Journal of the Science of Food and Agriculture

Line 373:  Change “Food Rev. Int.” to “Food Reviews International

Line 411:  Correct to read “…2019, 40(4). 432-444…”

Author Response

Thank you for all your contributions. Attached is a document highlighting the points that have been addressed in red. I particularly appreciate your review of the document.

While I didn’t find two observations in the literature, I went through everything again and made additional corrections. Thank you for your input!
